# NEK6 Regulates Redox Balance and DNA Damage Response in DU-145 Prostate Cancer Cells

**DOI:** 10.3390/cells12020256

**Published:** 2023-01-07

**Authors:** Isadora Carolina Betim Pavan, Fernanda Luisa Basei, Matheus Brandemarte Severino, Ivan Rosa e Silva, Luidy Kazuo Issayama, Mariana Camargo Silva Mancini, Mariana Marcela Góis, Luiz Guilherme Salvino da Silva, Rosangela Maria Neves Bezerra, Fernando Moreira Simabuco, Jörg Kobarg

**Affiliations:** 1Laboratory of Signal Mechanisms, School of Pharmaceutical Sciences (FCF), University of Campinas (UNICAMP), Campinas 13083-871, Brazil; 2Multidisciplinary Laboratory of Food and Health, School of Applied Sciences (FCA), University of Campinas (UNICAMP), Limeira 13484-350, Brazil; 3Department of Biochemistry, Federal University of São Paulo, São Paulo 04044-020, Brazil

**Keywords:** NEK6, CRISPR/Cas9, ROS, DDR, apoptosis

## Abstract

NEK6 is a central kinase in developing castration-resistant prostate cancer (CRPC). However, the pathways regulated by NEK6 in CRPC are still unclear. Cancer cells have high reactive oxygen species (ROS) levels and easily adapt to this circumstance and avoid cell death by increasing antioxidant defenses. We knocked out the NEK6 gene and evaluated the redox state and DNA damage response in DU-145 cells. The knockout of NEK6 decreases the clonogenic capacity, proliferation, cell viability, and mitochondrial activity. Targeting the NEK6 gene increases the level of intracellular ROS; decreases the expression of antioxidant defenses (SOD1, SOD2, and PRDX3); increases JNK phosphorylation, a stress-responsive kinase; and increases DNA damage markers (p-ATM and γH2AX). The exogenous overexpression of NEK6 also increases the expression of these same antioxidant defenses and decreases γH2AX. The depletion of NEK6 also induces cell death by apoptosis and reduces the antiapoptotic Bcl-2 protein. NEK6-lacking cells have more sensitivity to cisplatin. Additionally, NEK6 regulates the nuclear localization of NF-κB2, suggesting NEK6 may regulate NF-κB2 activity. Therefore, NEK6 alters the redox balance, regulates the expression of antioxidant proteins and DNA damage, and its absence induces the death of DU-145 cells. NEK6 inhibition may be a new strategy for CRPC therapy.

## 1. Introduction

Prostate cancer is the second most frequent cancer diagnosis in men and the fifth leading cause of death worldwide [1]. Androgen deprivation therapy is the standard treatment for prostate cancer. Unfortunately, although nearly all patients respond to treatment, most of these individuals will eventually progress to a fatal stage of the disease called castration-resistant prostate cancer (CRPC) [2].

NIMA (Never In Mitosis, gene A)-related kinase-6 (NEK6) belongs to a protein kinase superfamily composed of 11 members of NIMA-related kinases [3]. Although NEK proteins are poorly studied, they are known to be involved in cell cycle regulation [4], primary cilium function [5,6,7], and DNA damage response [8]. Additionally, a few recent studies have emerged exploring the relationship between NEKs and mitochondrial activity [9,10] and also emphasized the family of NEKs as biomarkers of several types of cancer [11].

NEK6 is a 313 amino acid serine/threonine kinase encoded in humans by the *NEK6* gene located at chromosome 9 [9,10]. Regarding its known functions, NEK6 participates in mitotic spindle kinetochore fiber formation, metaphase-anaphase transition, cytokinesis, and the checkpoint [8]. NEK6 is also involved in liver, breast, colorectal, gastric, and retinoblastoma [12,13,14,15,16,17,18,19]. NEK6 inhibition does not alter the cell cycle of normal cells, only cancer cells, indicating that the inhibition of NEK6 may provide therapeutic advantages in cancer treatment [20]. A recent study emphasized that NEK6 is an executable target in cancer [21]. A high-throughput genetic screen designed to establish new kinases involved in CRPC [22] identified the NEK6 protein as a central kinase that mediates androgen-independent tumor growth and determined that NEK6 is aberrantly expressed in human prostate cancer and several prostate cancer cell lines. The kinase activity of NEK6 appears to be directly involved with cell survival in CRPC [22]. However, it is still unclear what mechanisms and pathways NEK6 may participate in CRPC.

Reactive oxygen species (ROS) are a class of highly reactive, oxygen-containing molecules involved in survival signaling [23,24]. An excess or deficient level of ROS improves the chances of cell death or inhibition of cell growth through mediating ROS-dependent signaling, representing a novel anticancer therapeutic strategy based on ROS regulation [25]. Usually, cancer cells have higher levels of ROS compared to normal cells [26]. However, they suffer an adaptation of increased antioxidant capacity to maintain nonlethal ROS levels, then avoid cancer cell death [27]. Therefore, targeting the ROS signaling pathways and redox mechanisms in cancer progression are new potential approaches to cancer therapy. Additionally, the association of ROS inductors with conventional therapy (chemotherapy or radiotherapy) elevates the ROS levels above the suitable threshold, allowing a better efficiency and specificity to kill cancer cells [27,28,29]. ROS can also induce DNA damage and activate the DNA damage response (DDR), which culminates in DNA repair, cell cycle arrest, and cell death [30].

NEK6 roles are poorly studied, except for its well-characterized participation in the cell cycle [31,32]. However, only recently, the literature has shown its relevance as a possible potential therapeutic target in CRPC [9,21]. Thus, understanding the mechanisms that NEK6 regulates in prostate cancer models becomes essential for the design of NEK6 inhibitors. For these reasons, we targeted NEK6 depletion using the CRISPR/Cas9 system and evaluated the survival pathways NEK6 may regulate in DU-145 cells. We found that NEK6 regulates the redox balance and DNA damage response, inducing cell death. Furthermore, we show that NEK6 depletion is sensitive to cisplatin treatment. Therefore, NEK6 is an attractive target for developing new anticancer drugs.

## 2. Materials and Methods

### 2.1. Cell Culture

DU-145 (human prostate cancer cell line) was cultivated in RPMI 1640 (Thermo Scientific, #11875093) medium supplemented with 10% fetal bovine serum (FBS, #12657029) and 1% penicillin/streptomycin (Gibco, #15140-122). Cells were maintained at 37 °C in a humidified atmosphere containing 5% carbon dioxide. Cells were used for experiments within 10-25 passages from thawing.

### 2.2. NEK6 Knockout (NEK6-KO) Generation Using CRISPR-Cas9 System

To generate the NEK6-KO in DU-145 cells, two sgRNA sequences were designed using CRISPOR [33]. First, we chose guides that have high specificity and efficiency scores, which criteria were determined by Doench, Hsu, and colleagues [34,35]. One selected sgRNA targets exon 3 of the *NEK6* gene, called 83 forward sgRNA (sg83), sequence: 5′ AGGCCGAGGACAGTTCAGCG 3′. The other sgRNA targets exon 7 of the *NEK6* gene, named sgRNA 56 reverse (sg56), sequence: 5′ TGAAGAAGCGGCCCAGACCA 3′. Oligos containing the sequence were cloned into the PX459 vector (SpCas9(BB)-2A-Puro V2.0, Addgene, #62988); transfected in DU-145 cells using 2500 ng DNA, lipofectamine (Thermo Scientific, #18324012), and plus reagent (Thermo Scientific, #11514015); and cultured with 1 µg/mL puromycin for 72 h for select positive transfected cells. An empty PX459 vector was used as a mock control of transfection. Positively transfected cells were then isolated into 96-well plates by seeding 0.5 cells in 100 µL of medium per well using serial dilutions. This protocol was performed as described by Ran and colleagues [36]. The resulting monoclonal cultures were screened by Western blot for the loss of NEK6 expression using a NEK6 antibody (Santa Cruz, sc-50752). Cells transfected with the empty PX459 vector were also isolated in 96-well plates and used as mock control in the following experiments. To evaluate the indels generated by the CRISPR-Cas9 system, the targeted genomic region for NEK6 was amplified by PCR from genomic DNA and sequenced. The indels were characterized by cloning the amplicons in the pGEM-T vector (Promega, #A3600), followed by Sanger-type sequencing and detection of up to four different chromatograms (Appendix A).

### 2.3. DNA Genomic Extraction, Conventional PCR and RT-qPCR

Genomic DNA was extracted using the PureLink™ Genomic DNA Mini Kit (Invitrogen; Thermo Fisher Scientific, Inc., Waltham, MA,USA, K182001). The PCR reaction was performed following the manufacturer’s instructions (Invitrogen™, Waltham, MA,USA, #10966018) using 10 mM dNTP, 10× buffer, a mix of 10 mM primers each, 50 mM MgCl_2_, 500 ng of genomic DNA, 1.5 U of Platinum Taq DNA Polymerase^®^, and ultrapure water. The reaction was carried out under the following conditions: initial denaturation at 94 °C for 2 min, followed by 35 cycles with 3 steps: denaturation at 94 °C for 30 s, annealing at 55 °C for 30 s, and extension at 72 °C (1 min per kB of the gene). The primers used for genomic screening were:NEK6 genomic 83fwdF 5′ CAGCAGAGTCCCTCCTTCACCTTAGAG 3′;NEK6 genomic 83fwdR 5′ GATGGGTGGACATGGTATGAACCTCAG 3′;NEK6 genomic 56revF 5′ ACGTAGGCTGCTTCATGGAC 3′;NEK6 genomic 56revR 5′ GCCACAGCTGATTCCCTTCT 3′.

For real-time PCR, RNA was extracted from DU145 cells (WT and NEK6-KO) using TRIzol^®^ (Invitrogen™, #15596018). An amount of 2000 ng of RNA was used to synthesize cDNA using the High-Capacity cDNA Reverse Transcription kit (Thermo Fisher Scientific, Inc., #4368814). SYBR Green PCR Master Mix (Applied Biosystems, Waltham, MA,USA, #1725121) was used to perform the RT-qPCR. The quantification was measured according to the comparative threshold (Ct) cycle method using β-actin as a housekeeping gene. Samples, in triplicate, were arranged in a 96-well plate (MicroAmp, Applied Biosystems, #4306737) for amplification and were run in the Step One Plus Real-Time PCR System (Applied Biosystems). The primers used for RT-qPCR were:SOD1_Forward: 5′ GTTTCCGTTGCAGTCCTCG 3′;SOD1_Reverse: 5′ GGTCCATTACTTTCCTTCTGCTC 3′;SOD2_Forward: 5′AAGGAACGGGGACACTTACAAA 3′;SOD2_Reverse: 5′AGCAGTGGAATAAGGCCTGTTG 3′;PRDX3_Forward: 5′ GCCACATGAACATCGCACTCTTG 3′;PRDX3_Reverse: 5′ ACTGGGAGATCGTTGACGCTCA 3′;β-actin_Forward: 5′ GCCGCCAGCTCACCAT 3′;β-actin_Reverse: 5′ CCACGATGGAGGGGAAGAC 3′.

### 2.4. Cell Treatment

DU-145 cells (WT and NEK6-KO) were plated at a density of 2 × 10^5^ cells/well in 6-well plates. On the next day, the cells were treated with cisplatin (Calbiochem, CAS 15663-27-1) at 30 µM for 24 h. Western blotting was performed to analyze the expression of antioxidant proteins, DNA damage, and antiapoptotic proteins.

### 2.5. Colony Formation Assay

For the colony formation assay, cells were seeded in triplicate in 60 mm dishes (5 × 10^2^ cells/dishes). Cells were incubated for 8 days at 37 °C and stained with violet crystal solution (0.05% violet crystal *w/v*, 1% formaldehyde, 1% PBS, 1% methanol, and deionized water) for 20 min at room temperature. The number of colonies was manually quantified. Demonstrative images of the colonies were obtained under an optical microscope (Optika Italy) and using Optika Proview software.

### 2.6. Measurement of Mitochondrial Membrane Potential (ΔΨm)

Tetramethylrhodamine, ethyl ester (TMRE, Invitrogen™, #T669), is a red–orange fluorescent used to measure ΔΨm and selectively stain mitochondria. DU-145 cells (WT and NEK6-KO) were plated at a density of 1 × 10^4^ in a black 96-well microplate and incubated overnight in an incubator. TMRE treatment at 500 nM was performed for 30 min at 37 °C in a 5% CO_2_ incubator. The medium was removed, and in each well was added 100 μL of PBS. The fluorescence signal was analyzed at 549/575 nm in a microplate reader (Epoch, Biotek).

### 2.7. Proliferation and MTT Assay

DU-145 cells (WT and NEK6-KO) were plated at a density of 8 × 10^3^ cells/well in 96-well plates in triplicate. The proliferation was measured 4, 24, 48, and 72 h after seeding by the MTT assay. To evaluate cell viability, DU-145 cells (WT and NEK6-KO) were plated at a density of 8 × 10^3^ cells/well in 96-well plates in triplicate. The cells reached confluence in 3 days, and only then was the viability measured by adding 10 μL of 12 mM MTT (Invitrogen™, M6494) to each well and incubating for 2 h at 37 °C. The formazan crystals were solubilized in HCl and isopropanol solution for 15 min at 37 °C. The optical density was measured at 570 nm.

### 2.8. ROS Detection

The total amount of ROS present in cells was measured using dihydroethidium (DHE, Invitrogen™, D11347). Cells were trypsinized and centrifugated, and the pellets were resuspended in 500 µL of medium with DHE (5 µM) and incubated for 20 min at 37 °C. The cells were analyzed in BD Accuri C6^TM^ flow cytometry, with the acquisition of 5000 events. The fluorescence of DHE was analyzed in the FL2-A channel.

### 2.9. NEK6 Overexpression

DU-145 cells (WT and NEK6-KO) were plated at a density of 2 × 10^5^ cells/well in 6-well plates, and at 80% confluency, the cells were transfected using Lipofectamine 3000 (Invitrogen™, L3000001), following the manufacturer’s instructions. An amount of 2500 ng of pFLAG-NEK6 plasmid was used to overexpress NEK6 in cells. The same DNA mass of pFLAG-Ø, an empty vector, was used as the control of the assay. In the immunofluorescence assay, an amount of 400 ng of GFP-NEK6 plasmid was used to overexpress NEK6 in NEK6-KO cells.

### 2.10. Apoptosis Assay

Cell death was measured using the FITC Annexin V Apoptosis Detection kit (BD Pharmingen™, #556547). DU-145 cells (WT and NEK6-KO) were plated at a density of 2 × 10^5^ cells/well in 6-well plates and kept for 24 h at 37 °C in the incubator. The cells were washed with PBS, trypsinized, and 1×10^6^ cells were centrifuged for 150× *g* for 5 min, rewashed with PBS, centrifuged, and resuspended in 100 μL 1× Binding buffer. A small amount (3 μL) of FITC-Annexin V and 3 μL of propidium iodide (PI) were added to the cells, mixed, and incubated for 15 min at room temperature in the dark. An additional 100 μL of 1× Binding buffer was added to each sample and analyzed within 1 h. Controls were performed by staining with annexin only, another one with PI only, and another one in the absence of both reagents. The cells were analyzed in BD Accuri C6^TM^ flow cytometry, with the acquisition of 5000 events.

### 2.11. Subcellular Fractionation

Cells were trypsinized, centrifuged 150× *g* for 5 min, washed with PBS, and centrifuged again. The cell pellet was resuspended in a cytoplasmic extraction buffer (10 mM HEPES, 60 mM KCl, 1 mM EDTA, 0.075% (*v*/*v*) NP-40, 1 mM DTT, and protease inhibitor cocktail); incubated for 3 min on ice; and centrifuged in 1300× *g* for 30 min. The supernatant (cytoplasmic fraction) was transferred to new tubes. The nuclear pellet was washed with cytoplasmic buffer five times and resuspended in nuclear extraction buffer (20 mM Tris-HCl, 420 mM NaCl, 1.5 mM MgCl_2_, 0.2 mM EDTA, 25% (*v*/*v*) glycerol, and protease inhibitor cocktail); incubated for 10 min on ice; and vortexed every 2–3 min. Cytoplasmic and nuclear fractions were centrifuged at 12,000× *g* for 10 min, and the supernatants were collected.

### 2.12. Western Blotting

The proteins were collected from DU-145 cells using a cell lysis buffer (50 mM Tris-Cl, pH 7.5, 150 mM NaCl, 1 mM EDTA, 1% Triton X-100, protease, and phosphatase inhibitor cocktail), and samples containing 30 µg of total protein were separated by SDS-PAGE. The gels were electrotransferred to 0.45 µm nitrocellulose membranes (Bio-Rad Laboratories, Inc.) and incubated for 1 h at RT with 5% nonfat powdered milk dissolved in TBS-Tween-20 (TBS-T; 50 mM Tris-HCl, pH 7.5, 150 mM NaCl, and 0.1% Tween-20). Membranes were incubated with primary antibodies overnight at 4 °C, washed three times with TBS-Tween-20, and incubated with secondary antibodies for 1 h at room temperature, followed by washing three times with TBS-Tween-20. Protein bands were visualized using the Pierce ECL Western Blotting substrate (Thermo Scientific, #32106) or Clarity Max Western ECL substrate (Bio-Rad, #1705063) in the ChemiDoc Imaging System (Bio-Rad Laboratories, Inc.), and densitometry was performed using ImageJ software v1.53. Primary antibodies PRDX3 (Abcam, Ab73349), p-JNK (Cell Signaling, #4668), JNK (Cell Signaling, #3708), SOD1 (Cell Signaling, #4266), SOD2 (Cell Signaling, #13141), Bcl-xL (Cell Signaling, #2764P), Bcl-2 (Cell Signaling, #2870P), ATM (Cell Signaling, #2873P), pATM (Cell Signaling, #5883P), γH2AX, (Cell Signaling, #9718), Lamin A/C (Bethyl, #A303-430A), GAPDH (Santa Cruz, sc-25778), NEK2 (Santa Cruz, sc-33167), NEK3 (Santa Cruz, sc-100402), NEK4 (Santa Cruz, sc-81332), NEK5 (Santa Cruz, sc-84527), NEK6 (Santa Cruz, sc-50752), NEK7 (Abcam, ab133514), NEK8 (Santa Cruz, sc-50760), NEK9 (Santa Cruz, sc-50763), NEK11 (Santa Cruz, sc-100429), NF-κB2 (Cell Signaling, #4882), vinculin (Abcam, Ab18058), and α-tubulin (Calbiochem, CP06). All antibodies were used at 1:2000 stoichiometry. Secondary antibodies: HRP-conjugated goat anti-mouse IgG (Sigma-Aldrich, AP308P, 1:2000), goat anti-rabbit IgG (Sigma-Aldrich, AP307P, 1:5000), and goat (Sigma-Aldrich, A5420, 1:5000).

### 2.13. Immunofluorescence Assay

The assay was performed as described by Pavan and colleagues [37]. DU-145 NEK6-KO cells were transfected with GFP-NEK6, and after 24 h, the cells were fixed in 3.7% formaldehyde for 20 min, washed three times with PBS, permeabilized with 0.5% Triton X-100 for 10 min, and blocked with PBS containing 0.2% Triton X-100 and 3% bovine serum albumin (BSA) for 30 min. The cells were incubated with the NF-κB2 primary antibody rabbit (Cell Signaling, #4882) overnight at 4 °C, washed three times with PBS, and incubated with the secondary antibody anti-rabbit Alexa Fluor 594 (Invitrogen, A-11012). The cells were further incubated with Hoechst diluted 1:5000 in PBS for 10 min for nuclei staining. Coverslips were finally mounted using Prolong (Invitrogen™, P36980).

### 2.14. Statistical and Biostatistical Analysis

GraphPad Prism 8.01 software (https://www.graphpad.com/, accessed on 8 November 2022) was used to perform a statistical analysis. Data were presented as the means and SD. The mean difference was tested by the Student’s unpaired *t*-test or one-way ANOVA, followed by Tukey’s or Dunnett’s post-test, considering * *p* < 0.05, ** *p* < 0.01, *** *p* < 0.001, and **** *p* < 0.0001 as statistically significant. The GEPIA platform was used for the correlation of gene expression in prostate adenocarcinoma (PRAD). Spearman’s correlation coefficient was calculated to measure the strength of the correlation between two genes. The correlation degree was classified according to Schober and colleagues [38].

## 3. Results

### 3.1. Generation of NEK6-KO in DU-145 Cells Using the CRISPR-Cas9 Gene-Editing System

We first generated *NEK6* gene knockout in DU-145 cells using the CRISPR/Cas9 system to evaluate the effects of the lack of the NEK6 protein. We designed two sgRNAs against exons 3 and 7 of the *NEK6* gene, named sgRNA 83 and sgRNA 56, respectively (Figure 1A). We used two sgRNA to avoid the nonspecific effects of the CRISPR/Cas9 system. The specificity and efficiency scores were important for choosing the most appropriate sgRNA. Therefore, four NEK6-KO DU-145 cell lines were generated, named 83.7, 83.14, 56.3, and 56.5. Genomic DNA PCR products from each NEK6-KO cell line were demonstrated (Figure 1B). It is possible to visualize 631 and 754 base pair amplicons for sgRNA 83 and 56, respectively. We also observed an increase of a few base pairs in the 56.5 NEK6-KO cell line, which generated a second higher band in the gel. The genomic DNA PCR products of the NEK6-KO cell lines were sequenced, and the chromatogram of each clone was obtained (Figure 1C). To characterize these indels, we cloned the genomic DNA PCR products into a pGEM-T vector and sequenced the mutated locus of several clones (Appendix A). For all NEK6-KO cell lines, we obtained indels that generated premature stop codons. In the specific case of the 56.5 cell line, we detected an addition of 91 base pairs that explains the second-highest band shown in Figure 1B. We also analyzed the protein expression of NEK6 by Western blotting (Figure 1D). As a result, no detection of the NEK6 protein was observed in the NEK6-KO cell lines. WT cells were obtained using the same selection and isolation steps as the NEK6-KO cells.

As part of the NEK6 knockout characterization, we evaluated the expression of other NEKs in NEK6 knockout cells (Appendix A). NEK6 knockout increased NEK7 and NEK11 and decreased NEK2 and NEK9 expression at the protein level. NEK7 is a protein that has high structural similarity to NEK6, sharing 86% amino acid sequence identity at the C-terminal and only 20% identity at the N-terminal regions [39,40]. Thus, an increase in NEK7 expression was expected to partially compensate for the NEK6 lack. Additionally, together, NEK2, NEK6, NEK7, and NEK9 contribute to the establishment of the microtubule-based mitotic spindle [41], while NEK11 has been implicated in the DNA damage response [42]. NEK1 and NEK10 were not evaluated due to technical issues with antibodies. These data suggest new regulations between the NEKs in the cell cycle and DNA damage response.

### 3.2. Targeted Deletion of NEK6 in DU-145 Cells Reduces Clonogenic Capacity, Cell Proliferation, and Mitochondrial Membrane Potential

Malignant tumor cells can survive and grow without their neighboring cells and anchorage to the extracellular matrix (ECM) [43]. Therefore, we evaluated the clonogenic potential of NEK6-KO cell lines in the colony formation assay (Figure 2A). NEK6-KO cells exhibited a reduction in clonogenic potential. We also observed a decrease in the cell spread in NEK6-KO compared to WT colonies (Figure 2B), which may be related to a reduction in migration capacity. Quantification of the colony numbers was also manually performed, and the statistical analysis was shown (Figure 2C). We evaluated the proliferation rates of WT and NEK6-KO cell lines for 4, 24, 48, and 72 h (Figure 2D). All NEK6-KO cell lines, at 48 and 72 h, showed a significant decrease in proliferation: for 48 h (WT vs. 83.7 **, WT vs. 83.14 ***, WT vs. 56.3 **, and WT vs. 56.5 ***) and for 72 h (WT vs. 83.7 ****, WT vs. 83.14 ****, WT vs. 56.3 ****, and WT vs. 56.5 ****), with * *p* < 0.05, ** *p* < 0.01, *** *p* < 0.001, and **** *p* < 0.0001. The cells were maintained in the culture until reaching 100% cell confluence, and the MTT assay was performed to evaluate the cell viability (Figure 2E). We found that NEK6-KO cells presented a lower viability than WT cells, suggesting that mitochondrial activity may be downregulated in NEK6-KO cells. We stained the cells with TMRE to measure the mitochondrial membrane potential (Figure 2F) to prove this hypothesis. NEK6-KO cells demonstrated a diminished mitochondrial membrane potential, suggesting that NEK6 depletion alters the mitochondrial activity and viability.

### 3.3. Modulation in NEK6 Expression Alters ROS Levels and Antioxidant Defenses in DU-145 Cells

One of the reasons that could lead to a decrease in cell viability and depolarization of the inner mitochondrial membrane is the production of reactive oxygen species (ROS) [44]. Functionally compromised mitochondria generate even more ROS, mainly superoxide anion (O_2_^−^) and hydrogen peroxide (H_2_O_2_) [44]. Thus, we evaluated the presence of ROS in WT and NEK6-KO cells by DHE labeling (Figure 3A). NEK6-KO showed increased ROS levels, which could explain a reduction in the NEK6-KO cell viability and increased mitochondrial membrane depolarization observed in Figure 2E,F. Increased ROS levels usually result from an imbalance between the production of oxidants and default in their elimination by the antioxidant defense system [45]. Several enzymes, such as superoxide dismutase (SOD), catalase (CAT), glutathione peroxidase (GP_X_), and peroxiredoxin (PRDX), act in defense against oxidative stress [46,47]. Superoxide dismutase (SOD) converts superoxide anion (O_2_^−^) to hydrogen peroxide (H_2_O_2_), and PRDXs catalyze the reduction of H_2_O_2_, alkyl hydroperoxides, and peroxynitrite to water, alcohol, and nitrite, respectively [47]. The GEPIA platform was used to measure the correlation between the antioxidant protein expression: Superoxide Dismutase 1 and 2 (SOD1 and SOD2) and peroxiredoxin 3 (PRDX3) with the NEK6 gene expression in prostate adenocarcinoma samples. Spearman’s correlation coefficient showed a moderated correlation between the gene expression of SOD2 and NEK6 and PRDX3 and NEK6 based on coefficients of 0.5 and 0.44, respectively (Figure 3B).

Therefore, we evaluated the SOD1, SOD2, and PRDX3 expression at mRNA levels in WT and NEK6-KO cells by RT-qPCR (Figure 3C). The expression of SOD2 was significantly diminished at transcriptional levels in NEK6-KO cells; however, we did not observed any statistical difference in the SOD1 and PRDX3 mRNA levels. SOD1, SOD2, and PRDX3 protein expression were also evaluated by Western blotting (Figure 3D). A significantly reduced protein expression of SOD2 and PRDX3 was observed in all NEK6-KO cell lines, and a reduction of SOD1 expression was noted in 83.14 and 56.5 NEK6-KO cells lines, suggesting that increased ROS levels may be related to a lack of antioxidant defenses, such as SOD1, SOD2, and PRXD3. ROS also promotes the activation of c-Jun N-terminal kinase (JNK), a protein activated by a wide range of cellular stresses [48]. Then, we also evaluated the phosphorylation levels of JNK (T183/Y185), which revealed a statistically significant increase in its phosphorylation in NEK6-KO cells (Figure 3D,E). We also overexpressed FLAG-NEK6 in DU-145 WT cells and evaluated the SOD1, SOD2, and PRDX3 protein levels (Figure 3F). These antioxidant proteins had a statistically significant higher protein expression in NEK6 overexpression DU-145 cells, revealing that NEK6 regulates the antioxidant system through the SOD1, SOD2, and PRDX3 levels (Figure 3G). Here, we suggest that NEK6-KO cells showed higher oxidative stress, since a NEK6 lack causes a reduction in antioxidant defenses, which may be involved with the decrease in cell viability and mitochondria activity.

### 3.4. Targeted Deletion of NEK6 Increases DNA Damage Markers in DU-145 Cells

ROS are well known as mediators of DNA damage. Ataxia–telangiectasia mutated (ATM) is a sensor kinase of a double-stranded break (DSB), followed by a signaling cascade that culminates in the phosphorylation of histone H2AX [30]. ATM is directly activated by oxidative stress, leading to its autophosphorylation in serine 1981 [49]. Since NEK6 is involved in the cell redox balance, we explored whether NEK6 could be related to the DNA damage response. We analyzed the phosphorylation levels of ATM and H2AX in WT and NEK6-KO cells (Figure 4A,B). Western blotting confirmed that the levels of ATM phosphorylation in serine 1981 and γH2AX were increased in NEK6-KO cells compared to WT cells, suggesting NEK6 is involved in DDR. We confirmed this result by overexpressing NEK6 in WT cells, revealing that NEK6-overexpressing cells presented lower levels of γH2AX (Figure 4C,D). Additionally, we performed the rescue assay by overexpressing NEK6 in all NEK6-KO cell lines and observed that H2AX recovered its phosphorylation (Figure 4E,F). These results indicated that NEK6 regulates DDR pathways, which may be related to the reduction in viability, proliferation, and alteration in the redox balance in NEK6-KO cells (Figure 2 and Figure 3).

### 3.5. Targeted Deletion of NEK6 Induces Death in DU-145 Cells

ROS can activate several signaling pathways that can trigger the process of cell death [25]. One of them is the activation of JNK kinase, leading to apoptosis [50]. We already demonstrated in Figure 3 that NEK6-KO cells presented higher levels of JNK phosphorylation. Furthermore, significant alterations in the DNA damage response, redox balance, and mitochondrial membrane potential were shown in NEK6-KO cells. For this reason, we hypothesized that the loss of NEK6 expression could lead to cellular apoptosis. Therefore, WT and NEK6-KO cells were stained with annexin and propidium iodide and analyzed in flow cytometry, and the parameters such as early and late apoptosis and live cells were analyzed (Figure 5). The 83.7, 56.3, and 56.5 NEK6-KO cell lines showed higher levels of early and late apoptosis and reduced live cells (Figure 5A–D). Bcl-2, which plays an antiapoptotic role by regulating mitochondrial outer membrane permeabilization, was drastically reduced in the NEK6-KO cell lines (Figure 5E,F). The results suggest that elevated ROS levels and subsequent events in NEK6-KO cells may lead to cell death.

### 3.6. Targeted Deletion of NEK6 Sensitizes DU-145 to Cisplatin through Impairment of Antioxidant Defenses and Increase of DNA Damage

Cancer cells usually preserve a high basal level of ROS and are vulnerable to further increased ROS levels, exceeding a certain defensive threshold. Accordingly, ROS modulation has arisen as an anticancer strategy with the synthesis of various ROS-inducing or -responsive agents that target cancer cells [25,51]. It is known that exposure to cisplatin increases intracellular ROS and DNA damage, leading to apoptosis [52,53]. Considering that NEK6-KO cells have elevated basal levels of ROS, we treated these cells with cisplatin, an agent that induces ROS and DNA damage (Figure 6). We observed that the antioxidant defenses (SOD1, SOD2, and PRDX3); apoptosis marker (Bcl-xL); and DNA damage response (γH2AX) were altered in WT and NEK6-KO cisplatin-treated cells. We further identified that WT cells treated with cisplatin reduce SOD1 and PRDX3 expression. However, the NEK6-KO cells had a pronounced reduction in SOD1 and PRDX3 expression in the presence of cisplatin compared to the WT cells. Additionally, WT cells treated with cisplatin increased the SOD2 expression, while NEK6-KO had the opposite effect, diminishing the SOD2 expression. Additionally, we also observed that cisplatin treatment in the NEK6-KO cells strongly reduced Bcl-xL expression and increased γH2AX in comparison to the WT cells. These results indicate that the lack of NEK6 sensitizes DU-145 cells to cisplatin through a significant reduction in antioxidant defenses, the induction of DNA damage, and apoptosis.

### 3.7. NEK6 May Be Involved in the NF-κB2 Translocation to the Nucleus

It is well known that NF-κB promotes cellular survival [54]. One important mechanism by which NF-κB regulates survival pathways is increasing the expression of antioxidant proteins in response to the ROS levels. Therefore, NF-κB attenuates the ROS levels, protecting cells from ROS-induced death [55]. SOD2 is a well-known NF-κB transcription target gene [56,57,58]. Considering that NEK6-KO cells showed increased ROS, lower levels of SOD2 expression at the mRNA and protein levels and NEK6 overexpression elevated the SOD2 expression, we hypothesized that NF-κB localization and activity could be related to NEK6 expression. To validate this hypothesis, we performed cellular fractionation, isolating the nucleus and cytosol from WT, 83.14, and 56.5 NEK6-KO cells (Figure 7A). We observed that the p52 NF-κB2 subunit was more localized to the nucleus of WT cells than NEK6-KO cells, suggesting that the transcriptional activity of NF-κB2 may be intensified in the presence of NEK6 expression. These data reinforce the hypothesis that NEK6 knockout cells may have a deficiency in the activation of NF-κB2, which could be causing lower levels of SOD2. We also overexpressed NEK6 using GFP-NEK6 plasmid in NEK6-KO (56.5) cells and investigated whether NEK6 could be involved in NF-κB2 nuclear translocation by immunofluorescence (Figure 7B). These data suggest that NEK6 mediates the nuclear translocation of NF-κB2. However, further experiments need to be done to investigate this axis.

## 4. Discussion

This paper shows that NEK6 is involved in CRPC by regulating the redox balance and DNA damage response. The NEK6 lack in DU-145 prostate cancer cells reduced the proliferation, viability, and mitochondrial membrane potential and induced apoptosis. The literature already emphasizes that the family of NEKs should be explored as interesting targets in cancer treatment [11,19]. The inhibition of NEK6 has been seen as an executable target in cancer, and inhibitors of this kinase have already been explored in other types of cancer [21,59]. For this reason, it is essential to understand the survival pathways that NEK6 regulates in CRPC, because NEK6 inhibitors may be used as a cancer therapy.

Oxidative stress may cause aggressiveness in most cancer types, including prostate cancer. A moderate increase in ROS induces cell proliferation, whereas excessive amounts of ROS promote cell death [25,60]. Cancer cells react to an increase in the ROS levels and survive in this oxidative stress ambient by inducing the transcription of antioxidant enzymes. Then, it is essential to understand these pathways for efficiently elaborate therapies that modify the ROS levels. ROS inductors seem to have the ability to induce apoptosis and inhibit tumor growth, cell migration, and invasive properties in several cancers [60,61]. Here, we showed that a NEK6 lack reduces the expression of SOD2 at the mRNA level and also SOD1, SOD2, and PRDX3 at the protein level. Additionally, NEK6 overexpression in DU-145 cells increases the same antioxidant proteins. Along with that, the lack of NEK6 generates more ROS in DU-145 cells. These data suggest that NEK6 regulates the redox balance through the modulation of antioxidant proteins. Additionally, SOD1, SOD2, and PRDX3 have extensive literature showing their pro-oncogenic roles and their involvement with chemotherapy resistance through oxidative stress [62,63,64,65,66,67,68,69,70,71]. The modulation of these antioxidant proteins reveals a new way for NEK6 regulating cell survival in CRPC.

SOD1 is an essential antioxidant enzyme that is widely distributed in the cell. Its main function is to catalyze the dismutation of O_2_^−^ into H_2_O_2_, protecting cells from oxidative stress [64]. A study has shown that a specific SOD1 inhibitor repressed cancer cell growth and promoted cancer cell cycle arrest and apoptosis by mediating the ROS changes in several cancers [62]. SOD1 has also been linked to chemotherapy resistance [63,65]. SOD1 was identified as overexpressed in prostate cancer xenograft animals resistant to mitoxantrone (MTX), an antineoplastic agent used in CRPC [63]. For future studies, we suggest that an eventual combination of NEK6 inhibitors with MTX may be an interesting strategy for reducing the resistance of cancer cells to the chemotherapeutic agent.

SOD2 is an enzyme that catalyzes the dismutation of O_2_^−^ into H_2_O_2_ in the mitochondria, causing a reduction of ROS. The role of SOD2 in cancer is controversial. Initially, SOD2 was considered a tumor suppressor gene based on its low levels in several types of cancer [72]. However, more recent studies revealed that low levels of SOD2 are an early event of tumors, and higher levels of SOD2 are associated with tumor progression and metastasis [73]. The mRNA and protein levels of SOD2 were found elevated in samples from patients with middle stage prostate cancer [73]. SOD2 has recently been shown to increase the GLUT-1 and glucose uptake, which is essential for prostate cancer cell survival [74]. Several transcription factors are shown to regulate SOD2 expression, such as Nuclear Factor-Kappa B (NF-κB) and Nuclear factor erythroid 2-related factor 2 (Nrf2), among others [75,76]. Since the absence of NEK6 has modulated the SOD2 expression at the mRNA and protein levels, we evaluated the nuclear content of p52 NF-κB2 in NEK6-overexpressing cells. We observed that NEK6 regulated the nuclear localization of p52 NF-κB2. We suggest NEK6 may also regulate survival pathways and SOD2 expression by modulating NF-κB signaling.

PRDX3 is a mitochondrial peroxidase, which plays a role in cell protection against oxidative stress by detoxifying peroxides [77]. PRDX3 is overexpressed in several types of cancer, including prostate cancer, protecting cells against apoptosis [68,69,70,71]. Additionally, PRDX3 has been found overexpressed in castration-resistant prostate cancer cells, which culminates in promoting cell survival by protecting them from oxidative stress [71]. For this reason, PRDX3 is a potential target for CRPC. A study by our group showed that NEK6 interacts with PRDX3 [40]. NEK6 may regulate the expression of PRDX3 through this previously established interaction.

ROS promote the phosphorylation and activation of JNK and DNA damage response, leading to apoptosis [48,78]. The phosphorylation and activation of ATM and JNK mediate the formation of yH2AX foci [79]. NEK6 knockout induces ROS generation, activates JNK, a stress-sensing kinase, and induces DNA damage. Considering the cellular effects of a NEK6 lack on DNA damage and apoptosis, we suggest that future studies can explore NEK6 inhibitors as potential inducers of synthetic lethality in prostate cancer.

Bcl-2 is located in the mitochondrial outer membrane (MOM) and plays important antiapoptotic roles [80]. The permeabilization of MOM releases proapoptotic factors such as the apoptosis-inducing factor (AIF) and cytochrome C from the mitochondria. AIF enters the nucleus and generates extensive DNA fragmentation, while cytochrome C in the cytosol can initiate the activation cascade of caspases. The antiapoptotic Bcl-2 contributes to MOM integrity and prevents the release of these proapoptotic factors from the mitochondria [81]. Additionally, Bcl-2 is critical for the survival of androgen-independent prostate cancer cells and also required for the progression of prostate cancer cells from an androgen-dependent to an androgen-independent growth stage [82]. Thus, we evaluated apoptosis and Bcl-2 expression in the WT and NEK6-KO cell lines. We revealed that NEK6-KO reduces Bcl-2 protein expression and increases apoptosis in DU-145 cells. We also found that NEK6-KO cells showed an increase in mitochondrial membrane depolarization, which may be related to a reduction in the Bcl-2 levels and induction of apoptosis. These results revealed novel cell survival mechanisms regulated by NEK6 in a cellular model of CRPC.

By exploring the modulation of antioxidant defenses in WT and NEK6-KO cells, we observed that SOD1, SOD2, and PRDX3 expression were significantly reduced in NEK6-KO cells treated with cisplatin when compared to WT cells, which may explain why NEK6-KO cells have a higher sensitivity to this drug. The increase in SOD2 expression may be a mechanism that triggers a resistance to chemotherapy [66,67]. The TNF-α-mediated upregulation of SOD2 is involved in cisplatin resistance in esophageal cancer. The upregulation of SOD2 by TNF-α was inhibited by blocking the NF-κB pathway, suggesting that SOD2 via the NF-κB signaling pathway contributes to the proliferation of esophageal cancer cells [67]. In our data, SOD2 expression was found elevated in WT cells treated with cisplatin, suggesting a resistance mechanism in WT cells. A reduction in SOD2 expression in NEK6-KO cisplatin-treated cells means that this possible resistance mechanism found in WT was eliminated in NEK6-KO cells. Data showing a lower expression of Bcl-xL in NEK6-KO cisplatin-treated cells highlight an induction of apoptosis in these cells compared to WT cells. Cisplatin treatment also increased γH2AX in NEK6-KO cells compared to WT, showing a NEK6 lack causes genomic instability and may be responsible for increased cell death in NEK6-KO cells. Thus, we suggest that a NEK6 lack sensitizes cells to cisplatin, since the treatment with cisplatin in these NEK6-deficient cells greatly diminished the antioxidant defenses (SOD1, SOD2, and PRDX3), which may culminate in excessive ROS levels and DNA damage. Targeting NEK6 in combination with other anticancer drugs that destabilize the redox balance may be an interesting strategy to increase the ROS levels above the tolerable levels for cells. Another point is that, in Figure 2F, we observed a decrease in the mitochondrial membrane potential in NEK6-KO cells, indicating a reduction in mitochondrial activity. Moreover, cisplatin impairs the electron transport chain function, increases the intracellular ROS levels, and affects the mitochondrial viability [83]. Then, cisplatin may be acting in synergism with the effects in mitochondria mediated by the NEK6 lack, sensitizing these cells to cisplatin.

We performed our experiments using only one cell line, which may be considered a limitation of our study. We believe that the role of NEK6 in other prostate cancer cell lines should be further investigated, but our present data point out an important role of NEK6 in cisplatin sensitivity and as a co-adjuvant therapy for prostate cancer treatment.

## 5. Conclusions

This study showed that NEK6 regulates the redox balance and DNA damage response in a model of a castration-resistant prostate cancer cell line. The regulation of antioxidant defenses by NEK6 may be interesting in strategies focused on exacerbating the ROS levels and generating DNA damage. A NEK6 lack reduced proliferation, viability, cell migration, and the mitochondrial potential membrane, while elevating the ROS levels and DNA damage-induced cell death. A NEK6 lack sensitizes DU-145 cells to cisplatin. These data revealed that NEK6 may be an important therapeutic target in CRPC. Exploring the effects of NEK6 inhibitors on CRPC, alone and together with other chemotherapeutic agents, may be relevant for a new treatment approach. Future studies should investigate the regulation of NF-κB2 by NEK6 and explore whether it is mediated by physical interactions, for example, phosphorylation, or indirectly by other interactors. The involvement of NEK6 in the mitochondrial dynamics can also be investigated, as we have seen that NEK6 alters the expression of several mitochondrial proteins.

## Figures and Tables

**Figure 1 cells-12-00256-f001:**
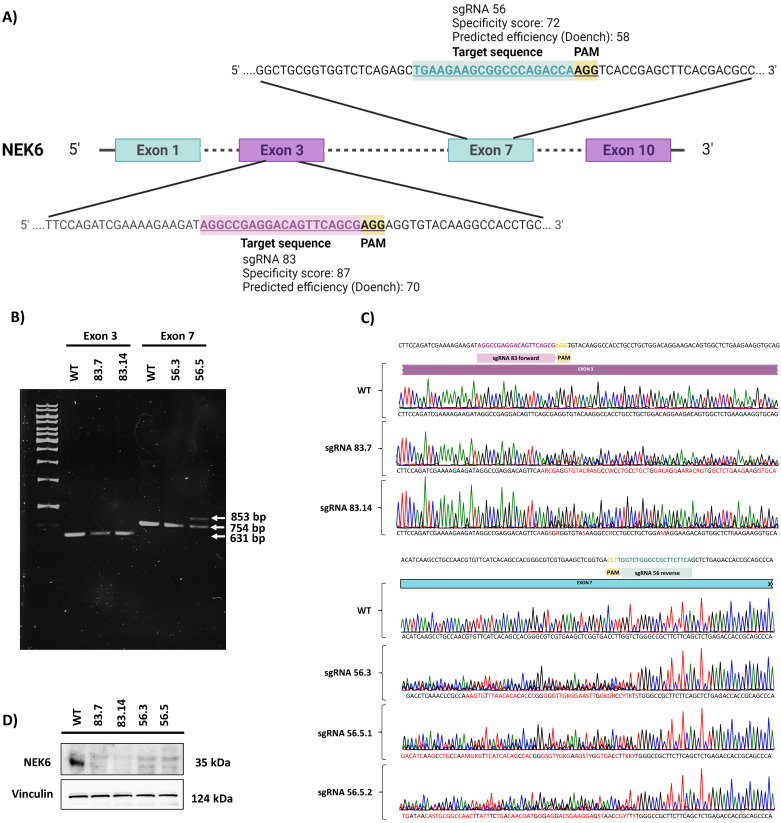
Generation and validation of NEK6-KO in DU-145 cells. (**A**) Gene targeting strategy for generating NEK6-KO in DU-145 cells. (**B**) PCR amplification of the mutated locus of NEK6-KO cell lines (83.7, 83.14, 56.3, and 56.5). (**C**) DNA sequence analysis showed the presence of indels adjacent to the PAM sequence in NEK6-KO cells. (**D**) Western blot analysis of NEK6 protein expression in WT and NEK6-KO cells.

**Figure 2 cells-12-00256-f002:**
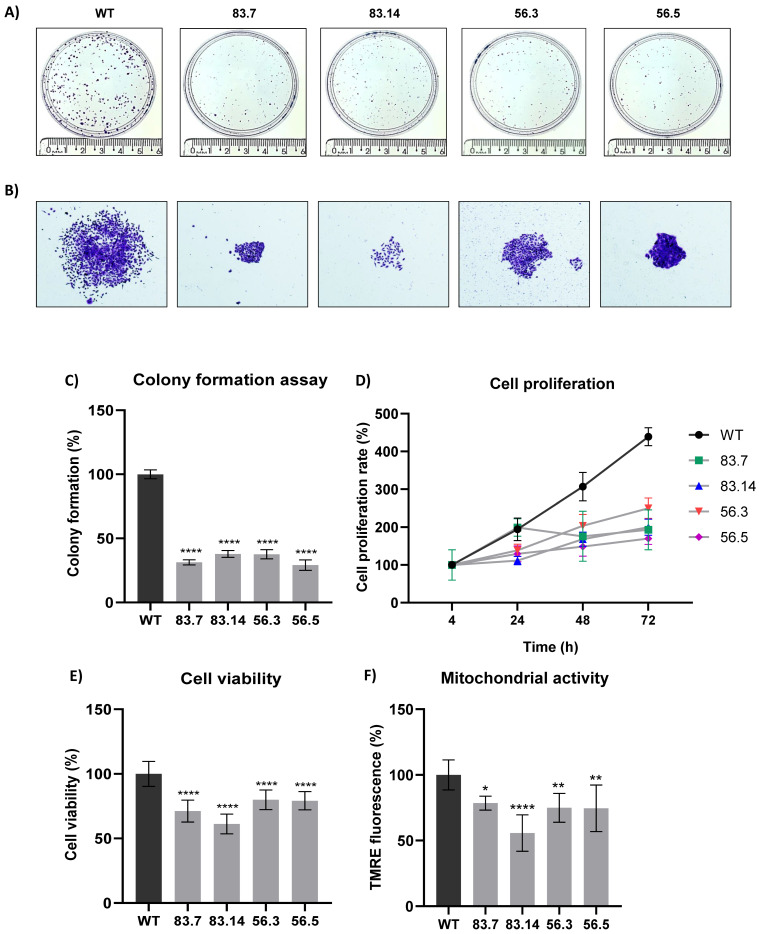
NEK6-KO reduces clonogenic capacity, cell proliferation, and mitochondrial membrane potential in DU-145 cells. (**A**) Deletion of the NEK6 gene significantly reduces the clonogenic potential in DU-145 cells. (**B**) Demonstrative images reveal the visual difference in colony size and its spread. (**C**) Quantification of colonies numbers obtained in WT and NEK6-KO cells. (**D**) Cells were evaluated for proliferation at 4, 24, 48, and 72 h after seeding, and the results showed all NEK6-KO clones significantly proliferated less than WT cells at 48 and 72 h. (**E**) Cells were maintained in an incubator (3 days) until reaching confluency. The MTT assay was performed to evaluate the viability of these cells. NEK6-KO showed a reduction in cell viability when compared to WT cells. (**F**) The target deletion of NEK6 in DU-145 cells also reduced the mitochondrial membrane potential using TMRE staining. * *p* < 0.05, ** *p* < 0.01 and **** *p* < 0.0001.

**Figure 3 cells-12-00256-f003:**
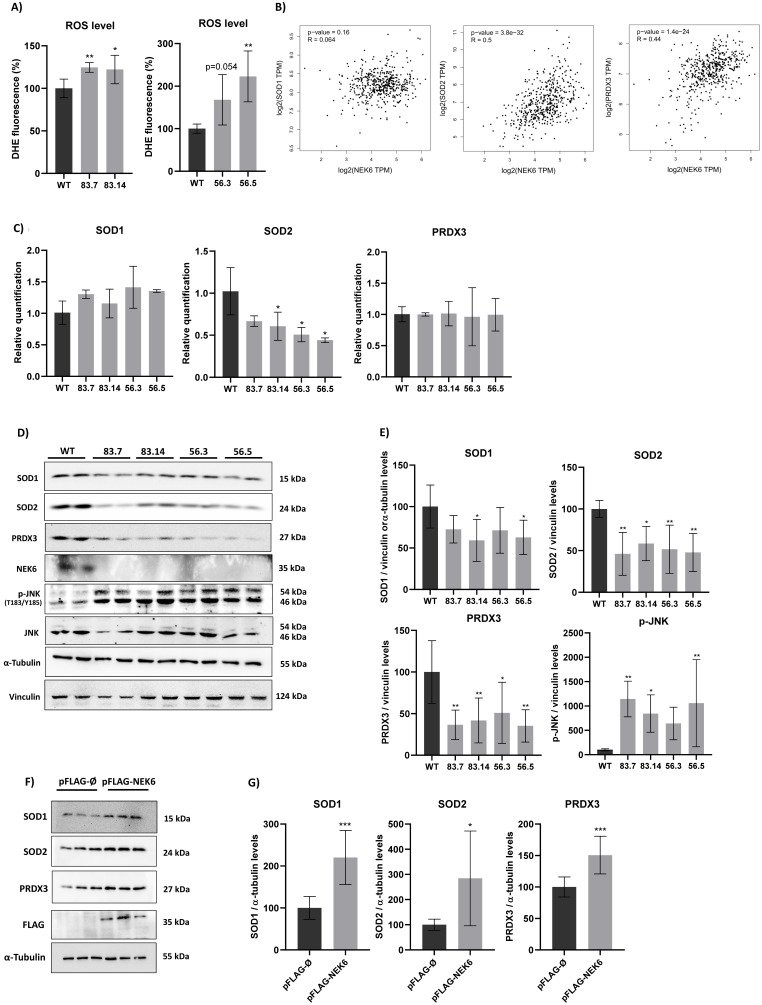
NEK6-KO increases ROS levels and its expression modulates antioxidant defenses in DU-145 cells. (**A**) DU-145 WT and NEK6-KO’s ROS levels were measured by DHE staining in flow cytometry. NEK6-KO showed an increase in ROS levels when compared with WT cells. (**B**) Correlation of the gene expression in prostate adenocarcinoma (PRAD) using the GEPIA platform. Spearman’s coefficient was used to measure the intensity of the correlation between genes. (**C**) mRNA levels of SOD1, SOD2, and PRDX3 were analyzed through RT-qPCR. A significant reduction in mRNA of SOD2 was observed. (**D**) The expression of antioxidant proteins (SOD1, SOD2, and PRDX3) was evaluated by Western blotting. Target deletion of the NEK6 gene reduces the expression of SOD1, SOD2, and PRDX3. Additionally, high levels of JNK phosphorylation were observed in NEK6-KO cells. (**E**) Quantification of antioxidant protein expression in WT and NEK6-KO cells. (**F**) NEK6 overexpression in DU-145 WT cells upregulated the SOD1, SOD2, and PRDX3 protein expression. (**G**) Quantification of antioxidant protein expression in FLAG-NEK6 expressing cells. * *p* < 0.05, ** *p* < 0.01 and *** *p* < 0.001.

**Figure 4 cells-12-00256-f004:**
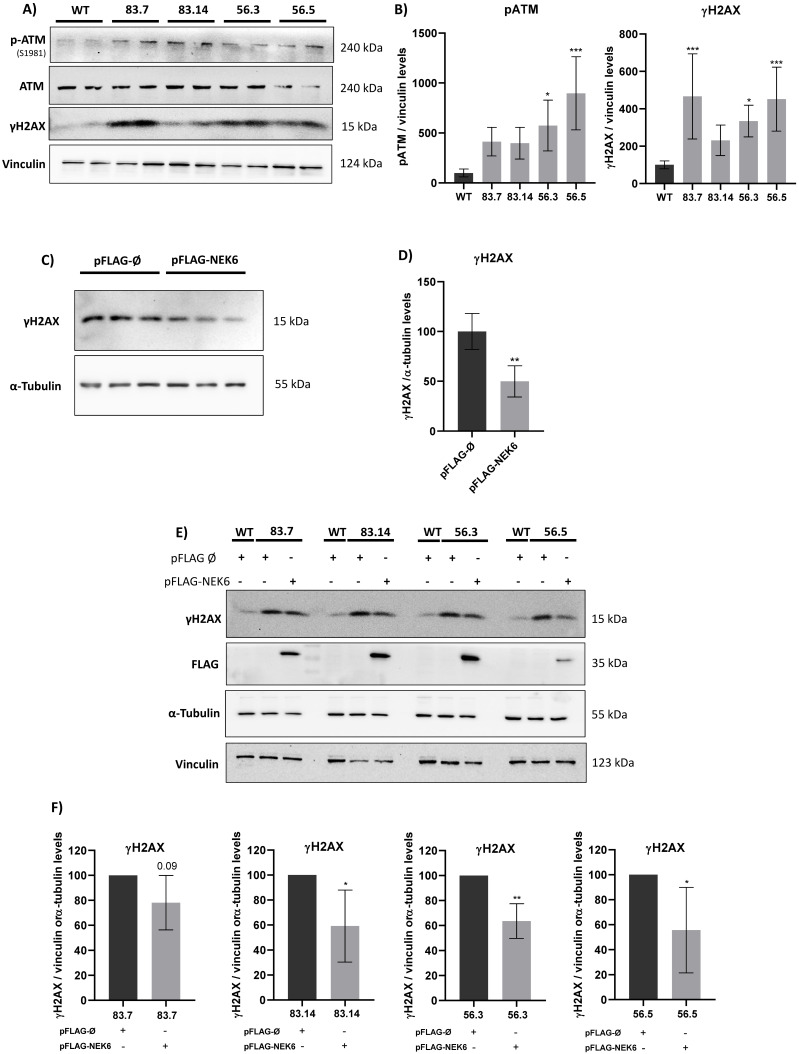
NEK6-KO increases DNA damage response in DU-145 cells. (**A**) NEK6-KO showed an increase in the phosphorylation of DNA damage markers, such as ATM and H2AX. (**B**) Quantification and statistical analyses of the results of Figure 4A. (**C**) NEK6 overexpression in DU-145 WT cells downregulated the phosphorylation of H2AX. (**D**) Quantification and statistical analyses of the results of Figure 4C. (**E**) NEK6 overexpression in NEK6-KO cells partially restores the H2AX phosphorylation levels. (**F**) Quantification and statistical analyses of the results of Figure 4E. * *p* < 0.05, ** *p* < 0.01 and *** *p* < 0.001.

**Figure 5 cells-12-00256-f005:**
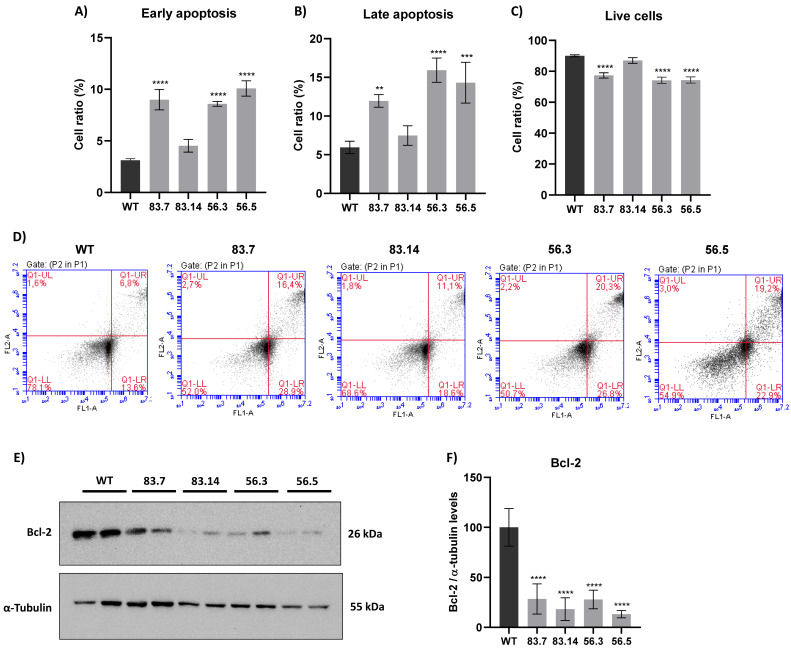
NEK6-KO induces cell death. (**A**) WT and NEK6-KO cells were stained with annexin V and propidium iodide to assess cell death by flow cytometry. NEK6 knockout cells (83.7, 56.3, and 56.5) showed a significant induction of early and (**B**) late cell death, (**C**) with the consequent reduction of live cells. (**D**) Representation of plots obtained by the flow cytometry analysis. (**E**) Bcl-2 expression was analyzed in NEK6-KO cell lines by Western blotting. (**F**) Statistical analysis of Bcl-2 expression. Bcl-2 expression was drastically reduced in NEK6-KO cells. ** *p* < 0.01, *** *p* < 0.001, and **** *p* < 0.0001.

**Figure 6 cells-12-00256-f006:**
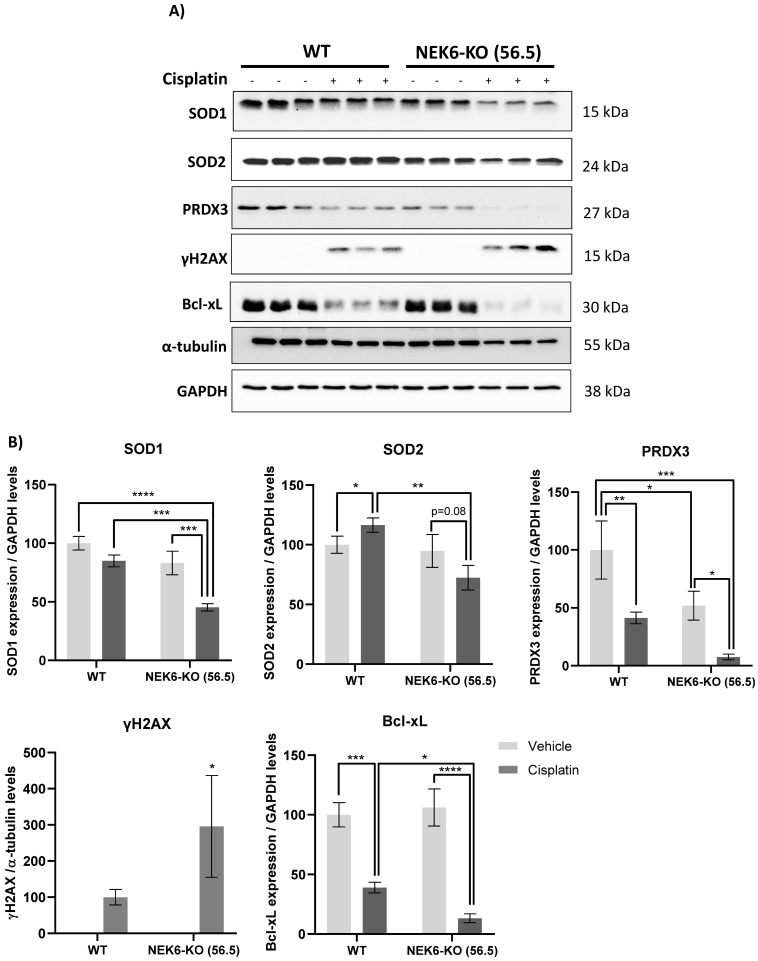
NEK6 depletion sensitizes DU-145 to cisplatin through the impairment of antioxidant defenses and increase of DNA damage. (**A**) WT and NEK6-KO (56.5) cells were treated with cisplatin 30 µM for 24 h. SOD1, SOD2, PRDX3, Bcl-xL expression, and γH2AX were evaluated by Western blotting. (**B**) Western blotting quantification was performed using ImageJ. * *p* < 0.05, ** *p* < 0.01, *** *p* < 0.001, and **** *p* < 0.0001.

**Figure 7 cells-12-00256-f007:**
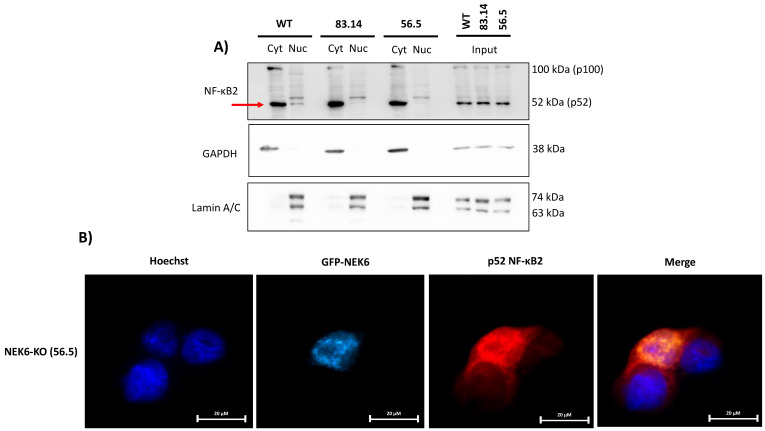
NEK6 is involved in the nuclear translocation of NF-κB2. (**A**) Nuclear (Nuc) and cytoplasmic (Cyt) fractions were obtained from WT and NEK6-KO (83.14 and 56.5) cells. GAPDH and Lamin A/C were used as the cytoplasmic and nuclear control, respectively. (**B**) NEK6-KO (56.5) cells were transfected with GFP-NEK6, and an immunofluorescence assay was performed to stain NF-κB2 in red. Hoechst 33342 was used for staining the nuclei.

## Data Availability

Not applicable.

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
