# Peer review of "NEK6 Regulates Redox Balance and DNA Damage Response in DU-145 Prostate Cancer Cells"

_cells, 2023, doi:10.3390/cells12020256_

Round 1
Reviewer 1 Report
The transition from androgen-dependent to androgen-independent is considered a key factor in PCa progression and resistance to hormonal therapy. In this manuscript, Pavan et al used the castration-resistant cell line DU-145 to demonstrate that the NEK6 kinase play an important in PCa transition from androgen-dependent to androgen-independent. Using CRISPR-Cas9 technology, the authors knocked down NEK6 expression in DU-145, and showed that inhibition of the kinase increases intracellular ROS production thereby inducing cell death. They have also shown that NEK6 deficient cells are more sensitive to cisplatin, a chemotherapeutic drug known for its apoptotic effects on several cancer cells. Based on their analysis, the authors concluded that NEK6 regulates the redox balance and DNA damage response in castration resistant prostate cancer (CRPC) cells, and that NEK6 inhibition may be a new strategy for CRPC therapy. The experiments are well-carried out and the results are convincing. A major drawback, however, is that the conclusions are drawn from one CRPC cell line. Each PCa cell line is like a “different world”. Thus, adding another CRPC cell line such as C4-B to the study and demonstrate the opposite in the parent LNCaP could lend more credence to the study and boost its suitability for publication.

Author Response
Answer: Dear reviewers of Cells, thank you for carefully reviewing the manuscript and for all the comments below. We appreciate all the suggestions pointed out, which have certainly improved the quality of our manuscript. We are resubmitting the manuscript with the modifications to address the questions presented. Please find below our answers to each question raised. We also performed a more accurate review of the English language used in this manuscript, promoting improvements in it. We thank you for the opportunity of resubmission and look forward to the decision for publication in Cells.
Reviewer 1:
Comments and Suggestions for Authors: The transition from androgen-dependent to androgen-independent is considered a key factor in PCa progression and resistance to hormonal therapy. In this manuscript, Pavan et al used the castration-resistant cell line DU-145 to demonstrate that the NEK6 kinase play an important in PCa transition from androgen-dependent to androgen-independent. Using CRISPR-Cas9 technology, the authors knocked down NEK6 expression in DU-145, and showed that inhibition of the kinase increases intracellular ROS production thereby inducing cell death. They have also shown that NEK6 deficient cells are more sensitive to cisplatin, a chemotherapeutic drug known for its apoptotic effects on several cancer cells. Based on their analysis, the authors concluded that NEK6 regulates the redox balance and DNA damage response in castration resistant prostate cancer (CRPC) cells, and that NEK6 inhibition may be a new strategy for CRPC therapy. The experiments are well-carried out and the results are convincing. A major drawback, however, is that the conclusions are drawn from one CRPC cell line. Each PCa cell line is like a “different world”. Thus, adding another CRPC cell line such as C4-B to the study and demonstrate the opposite in the parent LNCaP could lend more credence to the study and boost its suitability for publication.
Answer: Thank you for the suggestion. We completely agree that adding another prostate cancer cell line would give the study more credibility. However, the involvement of NEK6 in castration-resistant prostate cancer is recent, as well as its regulated signaling pathways are not well understood. In this study, we tried to explore new biological processes regulated by NEK6 that may justify its involvement in CRPC. We hope that this study will open up new possibilities for NEK6 to be investigated in other CRPC cell lines and also in other types of cancer. We recognize that this point raised is a limitation of the study, therefore, we emphasized that these results are restricted to DU-145 cells and it would be interesting to evaluate the regulation of oxidative stress and NEK6-mediated DNA damage in other CRPC cell lines, as well as other cancer types. We have added the following text in the discussion session: “We have performed our experiments using only one cell line, which may be considered a limitation of our study. We believe that the role of NEK6 in other prostate cancer cell lines should be further investigated, but our present data point out an important role of NEK6 in cisplatin sensitivity and as a coadjuvant therapy for prostate cancer treatment.”

Reviewer 2 Report
In this manuscript, Pavan et al. has studied the functional role of NEK6 in CRPC. With a NEK6-knock-out model, the authors found that NEK6-lacking cells reduced antioxidant proteins, increased ROS levels, and DNA damage, and became sensitive to cisplatin treatment and induced cell death. This MS provides a general insight into the interplay of NEK6 and related pathways and suggests that NEK6 inhibitors may be used as a cancer therapy. This is an interesting topic, and the authors did well on their experimental design and result presentation. The manuscript is generally well-written with informative figures. In general, I evaluate this as a good paper. I only have some minor revision suggestions. Their intention is to help the authors improve the manuscript further. They can be addressed through in-text revisions.
Ln 37-41. The logic here is a little bit confusing. Maybe something like, "Besides, a few recent studies have emerged exploring the relationship between NEKs and mitochondrial activity [REF] and also emphasized the family of NEKs as biomarkers of several types 41 of cancer [9]."
Ln 61. "cancer cells have higher levels of ROS compared to normal cells [better to insert a REF here]."
Ln 71. Removing "however" is better
Ln 99, 100. Is there a need to insert link for serial dilutions, western blot? Or this is required by the journal.
Ln 268. Although I don’t think this is an irrefutable must, but I think it is better to replace the Figure 1B with a better quality figure if the author do preserve the original DNA.
Ln 404. I think the figure legend of F is missing.
Ln 455. With the study of Cancer Res (2017) 77 (3): 753–765, there is no need to emphasize "the first time" here.
Ln 569. Instead of saying "Further studies need to understand the mechanisms 569 of these inhibitors and also the new roles of NEK6 in prostate cancer", the authors can list some detailed future points, like how to more solidly validate that NEK6 mediates the nuclear translocation of NF-κB2.

Author Response
Answer: Dear reviewers of Cells, thank you for carefully reviewing the manuscript and for all the comments below. We appreciate all the suggestions pointed out, which have certainly improved the quality of our manuscript. We are resubmitting the manuscript with the modifications to address the questions presented. Please find below our answers to each question raised. We also performed a more accurate review of the English language used in this manuscript, promoting improvements in it. We thank you for the opportunity of resubmission and look forward to the decision for publication in Cells.
Reviewer 2:
Open Review
Comments and Suggestions for Authors: In this manuscript, Pavan et al. has studied the functional role of NEK6 in CRPC. With a NEK6-knock-out model, the authors found that NEK6-lacking cells reduced antioxidant proteins, increased ROS levels, and DNA damage, and became sensitive to cisplatin treatment and induced cell death. This MS provides a general insight into the interplay of NEK6 and related pathways and suggests that NEK6 inhibitors may be used as a cancer therapy. This is an interesting topic, and the authors did well on their experimental design and result presentation. The manuscript is generally well-written with informative figures. In general, I evaluate this as a good paper. I only have some minor revision suggestions. Their intention is to help the authors improve the manuscript further. They can be addressed through in-text revisions.
Ln 37-41. The logic here is a little bit confusing. Maybe something like, "Besides, a few recent studies have emerged exploring the relationship between NEKs and mitochondrial activity [REF] and also emphasized the family of NEKs as biomarkers of several types 41 of cancer [9]."
Answer: The sentence has been rewritten.
Ln 61. "cancer cells have higher levels of ROS compared to normal cells [better to insert a REF here]."
Answer: A reference has been added in this sentence.
C.R. Reczek, N.S. Chandel, The Two Faces of Reactive Oxygen Species in Cancer, Annu. Rev. Cancer Biol. 1 (2017) 79–98. https://doi.org/10.1146/annurev-cancerbio-041916-065808.
Ln 71. Removing "however" is better
Answer: The word “however” has been removed.
Ln 99, 100. Is there a need to insert link for serial dilutions, western blot? Or this is required by the journal.
Answer: We checked the journal's instructions and did not find any instructions to quote serial dilution. However, we cited the study by Ran and colleagues, which explained the protocol in detail. Furthermore, we added information that 0.5 cells per well in a volume of 100 µL of medium are placed in the clone isolation process.
F.A. Ran, P.D. Hsu, J. Wright, V. Agarwala, D.A. Scott, F. Zhang, Genome engineering using the CRISPR-Cas9 system, Nat. Protoc. 8 (2013) 2281–2308. https://doi.org/10.1038/nprot.2013.143.
Ln 268. Although I don’t think this is an irrefutable must, but I think it is better to replace the Figure 1B with a better quality figure if the author do preserve the original DNA.
Answer: We believe that the gel was overexposed, so we corrected the overexposure of the bands by gently reducing the contrast, but without losing information from the original gel. Unfortunately, we do not have more of the original samples to repeat the analysis. Please see the attachment.
Ln 404. I think the figure legend of F is missing.
Answer: We have corrected the subtitle of figure 5.
Ln 455. With the study of Cancer Res (2017) 77 (3): 753–765, there is no need to emphasize "the first time" here.
Answer: We agree with your suggestion, so we removed this affirmation from the text.
Ln 569. Instead of saying "Further studies need to understand the mechanisms 569 of these inhibitors and also the new roles of NEK6 in prostate cancer", the authors can list some detailed future points, like how to more solidly validate that NEK6 mediates the nuclear translocation of NF-κB2.
Answer: The conclusion has been modified as suggested. We have added the following text in the conclusion: “Future studies should investigate the regulation of NF-κB2 by NEK6, and explore whether it is mediated by physical interaction, for example, phosphorylation, or indirectly by other interactors. The involvement of NEK6 in mitochondrial dynamics can also be investigated, as we have seen that NEK6 alters the expression of several mitochondrial proteins.”

Round 2
Reviewer 1 Report
The conclusions of this manuscript are based on results from one cell line. The authors have modified the manuscript to emphasized this point.